# Relationship of Postoperative Serum Neuro-Specific Enolase Levels with Postoperative Delirium Occurring after Microvascular Depression Surgery in Older Patients

**Tengxian Guo [1], Zhenxing Liu [2], Ji Qi [1,\*] and Zhen Wu [3,\*]**

[1] Department of Neurosurgery, Beijing Fengtai Hospital, Beijing 100070, China
[2] Department of Neurosurgery, Liaocheng Brain Hospital, Liaocheng People's Hospital, Liaocheng 252000, China
[3] Department of Neurosurgery, Beijing Tiantan Hospital, Capital Medical University, Beijing 100070, China
[\*] Correspondence: qiji76@aliyun.com (J.Q.); wuzhen1966@aliyun.com (Z.W.)

**Abstract:** There is a high incidence of postoperative delirium (POD) following microvascular decompression (MVD) surgery. Neuronal survival, differentiation, and neurite regeneration are regulated by neuro-specific enolase (NSE). Therefore, we investigated and assessed whether circulating NSE levels are related to POD after MVD surgery. We recruited a total of 209 patients and 209 age- and gender-matched healthy controls. A retrospective review of electronic medical records was conducted, and serum NSE levels were measured in the serum of patients before and after surgery, as well as the serum of controls. Patients were categorized according to the presence of POD. Postoperative patient serum levels of NSE were significantly higher compared to preoperative levels. Additionally, postoperative serum NSE levels were remarkably higher in POD patients than non-POD patients. In addition, there was no significant correlation between NSE levels and the type and severity of postoperative delirium. Age (OR = 1.153, 95% CI = 1.040–1.277, $p$ = 0.006), the levels of serum NSE (OR = 1.326, 95% CI = 1.177–1.494, $p$ < 0.001), and the levels of serum S100β (OR = 1.006, 95% CI = 1.000–1.012, $p$ = 0.048) were the three independent variables for predicting POD. A significant correlation existed between serum S100β levels and serum NSE levels ($t$ = 2.690, $p$ = 0.008). In terms of area under the precision–recall curve, the discriminatory ability of serum NSE levels (AUC = 0.876, 95% CI = 0.829–0.924, $p$ < 0.0001) was close to that of the serum S100β level (AUC = 0.879, 95% CI = 0.825–0.933, $p$ < 0.0001) and significantly higher than that of age (AUC = 0.813, 95% CI = 0.755–0.871, $p$ < 0.0001). Combining all three features produced a dramatic improvement over individual effects. The NSE level in serum was a stronger indicator of the likelihood of POD after MVD surgery in the older population. The clinical determination of this factor might be useful for distinguishing older patients at risk of POD after MVD surgery on the basis of their clinical findings.

**Keywords:** postoperative delirium; microvascular decompression; neuro-specific enolase

## 1. Introduction

During a postoperative recovery period, older patients may experience delirium, which is an acute state in cognition characterized by inattention, frequent shifts in consciousness levels, and/or disordered thinking [1–3]. The reported incidence rates range from 4% to 7% in patients who have undergone urologic surgery to nearly 35% to 65% in patients who have undergone hip fracture surgery, while they are as high as 32.4% among older patients in major head and neck surgery [3–7]. It is estimated that there are multiple risk factors for postoperative delirium (POD) among non-neurosurgical patients, whereas neurosurgical patients are frequently exposed to additional risks due to neurosurgical pathophysiology and neurosurgery-related damage to the brain, which might also be a contributing factor [8]. The impact of POD on neurological function has not yet been studied extensively in neurosurgery. For patients with primary cranial nerve disease,

including trigeminal neuralgia, hemifacial spasm, and various forms of neurovascular compression, microvascular decompression (MVD) has been scientifically proven to be the most effective treatment [9]. Specifically, 14.9–27.3% of patients with MVD experience POD after surgery [10]. While once thought to be transient and self-limiting, patients with POD have been associated with longer hospital stays, higher care costs, an increase in morbidity and mortality, and persistent perioperative neurocognitive disorders (PNDs) [11–14]. Identifying patients at high risk of POD is the best way to prevent it. Furthermore, early detection and effective intervention result in an excellent prognosis. Hence, it is essential to understand the risk factors for POD and make early predictions.

It is known that neuro-specific enolase (NSE) is an isoenzyme of enolase that is cell-specific. As a marker, it is capable of identifying not only all neuron types, but also all perineuronal cells or neuroendocrine cells [15]. There is evidence that it is a useful tool for measuring brain damage quantitatively and/or improving the outcome evaluation and diagnosis in seizures, ischemic stroke, intracerebral hemorrhage, and comatose patients after cardiopulmonary resuscitation for traumatic brain injury and cardiac arrest [16–22]. NSE has been shown to control neuronal survival, differentiation, and neurite regeneration by activating the PI3K/Akt and MAPK/ERK signaling pathways [23,24]. At present, circulating NSE levels are not examined in patients suffering from postoperative depression after neurosurgery. S100β is one of the members of the S100 gene group, which are proteins that can bind calcium. Previous studies have widely shown that higher S100β levels were found in patients with delirium than in patients without delirium, and S100β was proven to be the strongest independent marker in the pathogenesis of delirium [25–27]. Consequently, in this study, we recruited an older group of patients in need of MVD surgery, measured their preoperative and postoperative serum S100β levels, serum NSE levels, and relevant serum biochemical indicators, and further estimated the relationship between the development of POD and serum NSE levels.

## 2. Materials and Methods

### 2.1. Study Design, Setting, and Participants

An observational study was conducted at Beijing Tiantan Hospital, Capital Medical University. The recruitment period was from June 2020 to July 2021. Older (defined as age ≥65 years) healthy individuals were recruited as the controls, whereas older sufferers who received treatment for a hemifacial spasm (HFS) or trigeminal neuralgia (TN) were recruited as the cases. All patients included in this study had to meet the following criteria: (1) primary HFS or TN; (2) preoperative Mini-Mental State Examination (MMSE) score ≥24; (3) no prior specific pretreatment, such as Botox injection or radiofrequency ablation; (4) no previous psychiatric disorder such as delirium, dementia, or depressive illness; (5) no history of neurological disorders, such as trauma, intracranial tumor, or stroke; (6) no hematological disease, infectious disease, or serve diseases of the other systems, such as serious heart disease, metabolic syndrome, uremia, cirrhosis, or malignancy; and (7) no severe visual or auditory disorders. For the healthy control group, we collected the results of age-matched and gender-matched healthy individuals who performed their annual health check-ups at the hospital. Before participating in the study, all participants provided written informed consent, and ethics approval was granted by the institutional ethical committee of Beijing Tiantan Hospital. The study complied with the World Medical Association Code of Ethics (Declaration of Helsinki).

### 2.2. Data Source and Variables

The demographic characteristics of the patients, including age, gender, and disease spectrum (HFS or TN), as well as other clinical information relevant to this case, were registered. We also recorded the MMSE score, duration of anesthesia, time from symptoms to treatment, and hospitalization after surgery. Delirium was diagnosed and categorized by the Confusion Assessment Method (CAM) according to the Diagnostic and Statistical Manual of Mental Disorders, fifth edition (DSM-V, 2013) [28]. A postoperative delirium

assessment was conducted twice a day until the seventh day after surgery. The Memorial Delirium Assessment Scale (MDAS) [29] was used to assess the severity of delirium. After the patient's diagnosis of delirium was confirmed, the MDAS score was assessed every 2 days by two trained physicians. Triplicate results within 7 days after surgery were averaged, and the mean MDAS score was taken as the patient's MDAS score. PND was defined as the change in the MMSE score between baseline and 6 weeks post surgery, in line with the method applied in the study of Saxena et al. [29]. A neuropsychological assessment was carried out in a quiet setting on the ward. Investigators were trained to administer the tests by the psychologists responsible for the development of the test battery, and regular visits were made to ensure uniform collection of the data and administration of the assessment.

*2.3. Measurements*

A peripheral venous blood sample was collected on the first preoperative and postoperative days from 209 patients after an overnight fast. Serum NSE and S100β levels were determined by centrifuging samples at $1500\times g$, aliquoting, and freezing them at $-80\,^\circ$C. Commercially available enzyme-linked immunosorbent assay kits were used to quantify NSE and S100β in duplicate (Abcam ELISA, Shanghai, China, and Hengyuan ELISA, Shanghai, China, respectively) following the manufacturer's protocol. Analyses were conducted using the mean values of two measurements. Each determination was carried out by the same lab technician blinded to the clinical information.

*2.4. Statistical Methods*

In this study, SPSS 23.0 (IBM Corp., Chicago, IL, USA) and R studio 4.1.2(R Studio, Boston, MA, USA) were used to analyze the data. Values of $p < 0.05$ were considered significant. GraphPad Prism Software version 9.2 (GraphPad Software, San Diego, CA, USA) and R studio 4.1.2 (R Studio, Boston, MA, USA) were used to design the graphics. An analysis of data normality was conducted using the Kolmogorov–Smirnov test or the Shapiro–Wilk test. There was no normal distribution of continuous data; thus, median values were presented (interquartile ranges). Frequencies (percentages) were used to express categorical data. Continuous data were analyzed using the Mann–Whitney U test or the Wilcoxon rank-sum test, and categorical data between two groups were tested with the chi-square test or Fisher exact test. Spearman's correlation coefficient was used to calculate bivariate correlations. The variables that were confirmed to be significant were then combined into linear regression with multiple variables to examine the association between NSE levels and S100β levels. We assessed predictors of POD and adjusted for possible confounders using binary logistic regression. The variables that predicted POD with univariate logistic regression models were then analyzed using multivariate models. An odds ratio (OR) was estimated, and 95% confidence intervals (CIs) were calculated. On the basis of the areas under curves (AUCs) and 95% confidence intervals (CIs) obtained from the precision–recall (PR) curves, optimal prediction sensitivities and specificities were determined.

## 3. Results

*3.1. Study Population Characteristics*

According to the criteria for inclusion and exclusion, 81 male and 128 female patients were included in the study. For comparison with the patients, we collected 209 older healthy controls of similar ages and genders. Table 1 presents a summary of the clinical and laboratory characteristics of the patients. In total, 32 (15.32%) of the 209 recruitment patients were lost to follow-up, and 177 people had complete MMSE data. We classified delirium into three types: the hyperactive type ($n = 14$), the hypoactive type ($n = 26$), and the mixed type ($n = 6$). In addition, the severity of delirium was assessed using the MDAS (Memorial Delirium Assessment Scale, scores 0–30).

**Table 1.** Clinical characteristics and laboratory examinations and parameters correlated with serum NSE levels after MVD in older patients.

| Variable | Range | Number/Median (IQR) | Correlation Analysis | |
|---|---|---|---|---|
| | | | *r*-Value | *p*-Value |
| Gender (male/female) | | 81/128 | 0.048 | 0.490 |
| Age (years) | 65–92 | 73 (67–78) | 0.167 | 0.016 |
| Type of diseases | | | 0.024 | 0.735 |
| HFS | | 102 | | |
| TN | | 107 | | |
| Time from symptoms to treatment (months) | 6–121 | 21 (15–43.5) | 0.027 | 0.693 |
| Baseline MMSE score | 24–30 | 29 (27–30) | 0.261 | <0.001 |
| Duration of anesthesia (min) | 66–154 | 91 (82–97) | 0.215 | 0.002 |
| Hospitalization after surgery (days) | 7–15 | 8 (8–9) | 0.314 | <0.001 |
| CRP | 1–71 | 10 (9–16) | 0.157 | 0.023 |
| IL-6 | 2.00–325.00 | 28.80 (14.70–62.65) | 0.047 | 0.495 |
| TNF-a | 4.00–179.00 | 10.40 (7.00–12.05) | 0.045 | 0.514 |
| S100β | 310.00–1138.67 | 529.05 (419.52–600.48) | 0.189 | 0.006 |
| HbA1c | 4.4–9.4 | 6.00 (5.50–6.45) | 0.111 | 0.109 |
| Type of delirium | | | | |
| Hyperactive | | 14 | | |
| Hypoactive | | 26 | | |
| Mixed | | 6 | | |
| MDAS score (0–30) | | | | |
| <10 | | 5 | | |
| <20 | | 33 | | |
| ≤30 | | 8 | | |

Notes: Continuous data and categorical data are presented as medians (interquartile ranges, IQRs) and frequencies (percentages), respectively. Bivariate correlation was determined using Spearman's correlation coefficient.

### 3.2. Serum NSE Levels and Other Variables

The comparisons of serum NSE levels among different groups are demonstrated in Figure 1. No statistically significant difference was found between the serum NSE levels of patients and healthy controls before surgery. There was a significant increase in postoperative serum NSE levels compared with preoperative serum NSE levels in patients. The postoperative serum NSE levels of patients were consistently higher than those of healthy controls (Figure 1A). In addition, 46 (22.01%) of 209 patients were identified as POD in this study, and postoperative serum NSE levels were significantly higher in POD patients compared to non-POD patients (Figure 1B). However, no significant differences in postoperative serum NSE levels were seen between patients with different types of delirium occurring after surgery (Figure 1C). In addition, we used the MDAS scale to score the degree of postoperative delirium in patients. No significant difference in delirium severity was found between the different delirium types, and no significant correlation was found between delirium severity and postoperative serum NSE levels ($p = 0.1953$, Pearson's $r = -0.1945$) (Figure 1D,E).

An analysis of bivariate correlations showed that postoperative serum NSE levels were significantly associated with age, baseline MMSE, anesthesia duration, hospitalization following surgery, serum CRP levels, and serum S100β levels (Table 1). An analysis of a multivariate linear regression model revealed an independent association between serum NSE and serum S100β levels after incorporating the previously determined significant variables ($t = 2.690$, $p = 0.008$) (Figure 2).

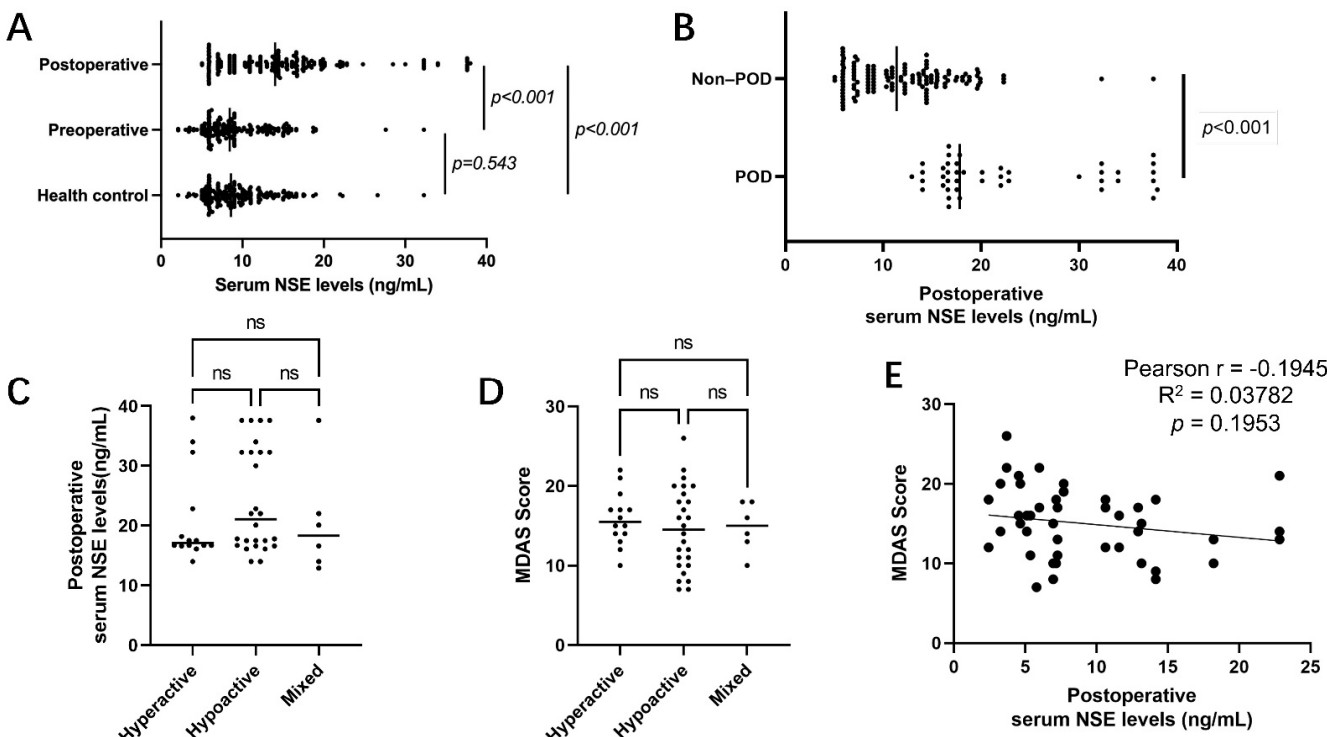

**Figure 1.** Comparisons of serum NSE levels between the healthy controls and the patients (**A**), between patients with POD and those without POD (**B**), and among patients with different types of delirium (**C**). There was no clear correlation between MDAS scores or the type of delirium and postoperative serum NSE levels (**D,E**). ns: no significance.

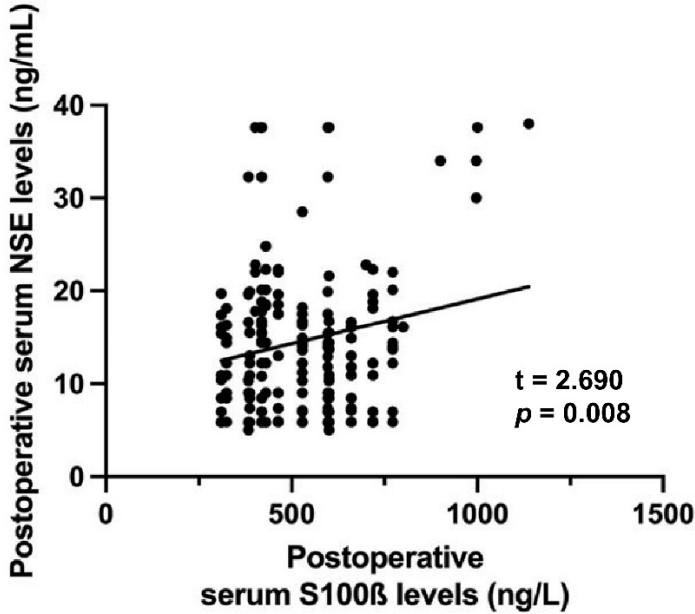

**Figure 2.** Relationship between postoperative serum NSE levels and serum S100β levels in older participants undergoing MVD surgery.

### 3.3. POD Prediction

As presented in Table 2, through univariate analysis, we could find that the POD patients were older and had lower baseline MMSE scores, longer anesthesia duration, more prolonged postoperative hospitalization, higher levels of serum CRP, higher levels of serum

NSE, and higher levels of serum S100β. Subsequently, the abovementioned parameters were introduced into the binary logistic regression model after being proven significant in univariate analysis, and it was presented that age (OR = 1.153, 95% CI = 1.040–1.277, $p$ = 0.006), serum NSE levels (OR = 1.326, 95% CI = 1.177–1.494, $p$ < 0.001), and serum S100β levels (OR = 1.006, 95% CI = 1.000–1.012, $p$ = 0.048) were the three independent predictors for POD. In addition, among the 177 patients with complete MMSE scores, a total of 45 patients (25.42%) experienced PND. The results revealed a statistically significant difference between the POD group and the non-POD group regarding the incidence of PND ($p$ = 0.026).

**Table 2.** Factors related to POD after MVD in older patients.

| Variable | POD | Non-POD | $p$-Value | OR (95% CI) | $p$-Value | OR (95% CI) |
|---|---|---|---|---|---|---|
| | | | Univariate Regression | | Multivariate Regression | |
| Gender | | | 0.495 | 0.958 (0.200–4.589) | | |
| Male | 20 | 61 | | | | |
| Female | 26 | 102 | | | | |
| Age (years) | 78.5 (76–82) | 70 (67–75) | <0.001 | 1.159 (1.044–1.288) | 0.006 | 1.153 (1.040–1.277) |
| Type of disease | | | 0.620 | 1.286 (0.237–6.975) | | |
| HFS | 24 | 78 | | | | |
| TN | 22 | 85 | | | | |
| Time from symptoms to treatment (months) | 20.5 (14.5–45.25) | 21 (15–42) | 0.945 | 1.011 (0.984–1.040) | | |
| Baseline MMSE score | 26 (25–27) | 29 (28–30) | <0.001 | 0.469 (0.289–0.763) | | |
| PND | 17(43) | 28(134) | 0.026 | 2.475(1.178–5.005) | | |
| Duration of anesthesia (min) | 98.5 (93–105.25) | 88 (80–95) | <0.001 | 1.121 (1.022–1.229) | | |
| Hospitalization after surgery | 9 (9–10) | 9 (8–9) | <0.001 | 3.620 (1.344–9.751) | | |
| HbA1c | 6.00 (5.50–6.43) | 6.00 (5.50–6.50) | 0.848 | 0.791 (0.363–1.722) | | |
| CRP | 19 (10.75–34) | 10 (9–13) | <0.001 | 1.143 (1.052–1.241) | | |
| IL-6 | 31.95 (19.13–67.35) | 27.30 (12.90–59.60) | 0.133 | 0.993 (0.973–1.014) | | |
| TNF-a | 10.80 (7.17–12.42) | 10.40 (7.00–11.70) | 0.218 | 0.955 (0.888–1.026) | | |
| S100β | 562.98 (419.52–771.90) | 463.67 (387.00–600.48) | 0.006 | 1.006 (1.000–1.012) | 0.048 | 1.006 (1.000–1.012) |
| NSE | 17.80 (16.60–32.30) | 11.20 (7.34–14.90) | <0.001 | 1.357 (1.184–1.554) | <0.001 | 1.326 (1.177–1.494) |

Notes: Continuous data are presented as medians (interquartile ranges). Categorical data are expressed as frequencies (percentages). Intergroup comparisons were performed using the Mann–Whitney U test for continuous data and the chi-square test or Fisher exact test for categorical data. The odds ratios (ORs) and 95% confidence intervals (CIs) are presented.

Figure 3 demonstrates the predictive value (PR curve) of age, serum NSE levels, and serum S100β levels, as well as their combinations, for the occurrence of POD. An optimal cutoff value of serum NSE levels (15.80 ng/mL) was selected according to the PR curve. The corresponding AUC was 0.876 (95% CI 0.829–0.924, $p$ < 0.0001). According to the AUC, the discriminatory ability of serum NSE levels was close to that of the serum S100β level (AUC = 0.879, 95% CI = 0.825–0.933, $p$ < 0.0001) and significantly higher than that of age (AUC = 0.813, 95% CI = 0.755–0.871, $p$ < 0.0001). When combining these three features to analyze the validity of prediction efficacy, the highest AUC value was obtained (AUC = 0.950, 95% CI = 0.917–0.982, $p$ < 0.0001).

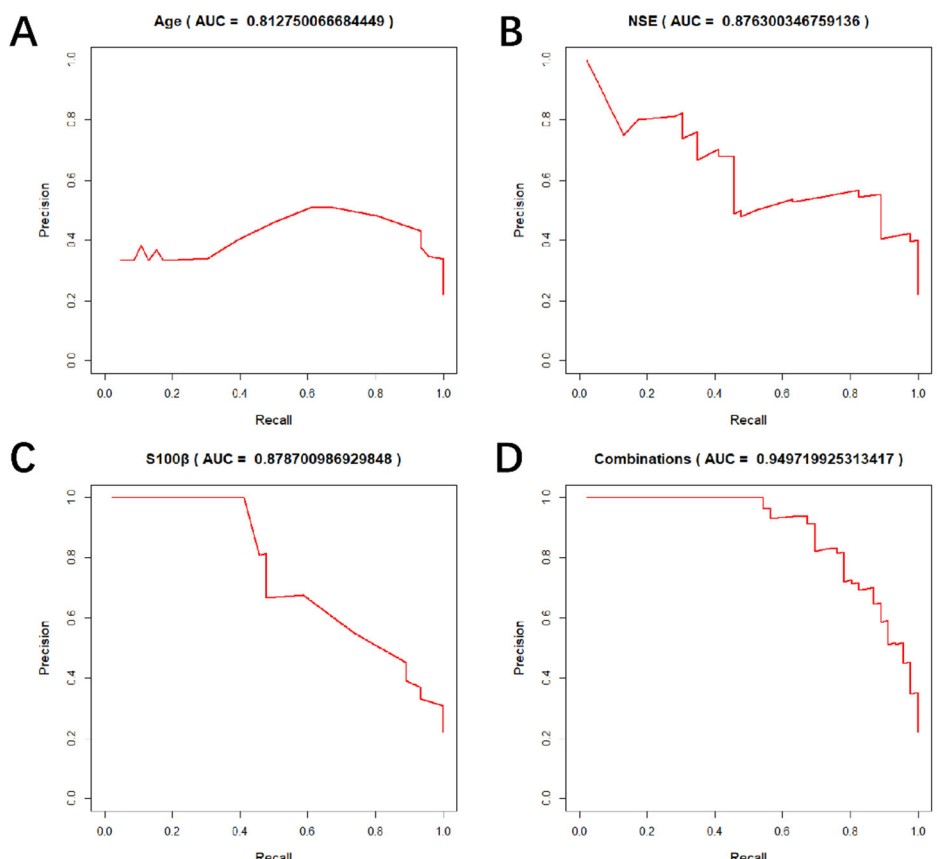

**Figure 3.** PR curves of age (**A**), postoperative serum NSE level (**B**), postoperative serum S100β level (**C**), and the combination of all three features (**D**) for discriminating POD in older participants undergoing MVD surgery.

## 4. Discussion

The present study indicated that more than 20% of MVD surgical patients experienced delirium postoperatively with a high occurrence rate of PND. The pathophysiology of PODs is unclear, and the cause of POD is also multifactorial. There are insufficient data available to make recommendations on the best way to reduce the incidence of POD using clinical therapy. In this study, there was an obvious increase in serum NSE levels following surgery in all patients compared with controls. A significant difference was also observed in the levels of postoperative serum NSE between patients who developed POD and those without. Furthermore, no significant differences in postoperative serum NSE levels were seen as a function of the type and severity of delirium, indicating that NSE is involved in the occurrence of delirium without affecting the type and severity. However, this result needs to be confirmed in a larger cohort, especially with more delirious patients. In the analysis of POD predictors, age, serum NSE levels, and serum S100β levels were identified as being the three independent factors. Moreover, serum NSE levels showed a strong predictive power for POD. In other words, serum levels of NSE following MVD surgery might serve as a potential biomarker to differentiate older participants who develop POD from those who do not.

A number of previous studies examined the role of NSE in glycolysis in neural and neuroendocrine tissues. The ability of serum NSE levels to predict functional neurological outcomes in stroke patients is a matter of recent interest, with some studies suggesting that NSE is useful in predicting functional outcomes [30–33]. Zaheer et al. studied 75 patients who suffered from acute ischemic stroke to estimate the relationship between levels of NSE at admission and infarct volume, the severity of the stroke, and early functional neurological outcome. The results showed that there was a significant negative correlation

between the Glasgow Coma Scale at presentation and concentration of NSE on day 1, and a positive correlation between the concentration of NSE on day 1 and early neurological outcome as evaluated by the modified Rankin scale on day 30 [31]. In the study, NSE serum levels were shown to function as a useful indicator of stroke severity and functional outcome in the early stages of ischemic stroke. Similarly, NSE levels have been shown to improve diagnosis and outcome evaluation across a variety of clinical situations, such as seizures, intracerebral hemorrhage, and comatose patients following cardiopulmonary resuscitation for cardiac arrest, as well as traumatic brain injury [18,19,21,22,34,35]. In addition, overexpressed NSE and S100β proteins have been reported to potentially damage the brain tissue [36]. Changes in the levels of NSE or S100β in serum or cerebral spinal fluid may reflect the damage of neurons and glial cells. Kimura et al. assessed the NSE before and 1 day after thoracic aortic surgery to show that it is a significant predictor of neurologic complications [37]. Zhang et al. investigated that the RNA interference (iRNA)-mediated silencing of the S100β gene could improve the recovery of nerve function while inhibiting the apoptosis of hippocampal cells in rats with ischemic stroke [38]. It was shown that S100β was the strongest independent marker in the pathogenesis of delirium [25–27]. In this study, the results exhibited that NSE was an independent predictor for POD. Additionally, the level of serum NSE correlated independently with that of serum S100β. Consequently, accumulating evidence implies that NSE could contribute to POD pathophysiology.

Delirium is an acute medical emergency, with psychiatric manifestations, which is seen across different treatment settings [39]. In our study, delirium was found to occur in 22.01% of postoperative MVD patients. Previous studies on the mechanisms of delirium development have proposed various hypotheses, such as neuroinflammation, neuronal aging, oxidative stress, neurotransmitter deficiency, neuroendocrine, circadian dysregulation, and network disconnection, including a variety of mechanisms, such as neurotransmitter synthesis and functional alterations, which may interact to jointly mediate the various cognitive and behavioral dysfunctions associated with delirium [40–42]. Several previous studies have suggested predictive indicators of postoperative delirium. Ross et al. discovered that abnormalities in resting-state EEG spectral power or TMS plasticity may indicate a subclinical risk for post-surgery delirium [43]. Lin et al. collected 740 patients, 83 of whom developed postoperative delirium, and examined preoperative biochemical markers of cerebrospinal fluid, suggesting that the preoperative α-syn protein level in cerebrospinal fluid can be a predictor of postoperative delirium [44]. The risk of POD can be attributed to many variables, but age continues to be an independent factor that is well established [8,45]. Moreover, according to Mu et al., delirium is also associated with operative time. Evidence suggests that, as the operation continues, higher cumulative anesthesia dosage results in a higher incidence of delirium [46]. Moreover, it is well known that inflammatory mediators, such as serum CRP, also play a crucial role in activating glial cells during POD development, which leads to inflammation and brain damage [47–49]. The present study supports these findings. In light of the fact that NSE has been reported in many previous studies of brain-related disorders, we investigated the clinical association between NSE and postoperative delirium. Our study found that patients with delirium secondary to MVD surgery had significantly elevated serum NSE levels. Neuron-specific enolase (NSE) is known to be a cell-specific isoenzyme of the glycolytic enzyme enolase and may be induced by intraoperative effects on nerve and brain tissue, suggesting that NSE could be an important marker for the development of postoperative delirium after MVD.

In addition, we discuss the mechanisms via which delirium is induced following MVD. Given the process of MVD from craniotomy to the intraoperative stage, in addition to neurological disturbance, there may be blood flow into the subarachnoid space and subsequent stimulation of brain tissue and neurovascularity. Previous studies have confirmed that, after subarachnoid hemorrhage, cell-free hemoglobin (CFH) may disrupt vascular endothelial cells, induce increased cerebrovascular permeability and vasogenic brain edema, promote inflammatory cell infiltration (especially mononuclear macrophages), and mediate multiple

pathological processes after SAH [50,51]. Pathologic heme exposure is a major mediator of SAH pathology, and CFH disruption of vascular endothelial cells may affect the expression of adropin. Adropin, a conserved peptide hormone highly expressed in the brain and liver, acts directly on endothelial cells to produce cytoprotective and vascular protective effects by stimulating nitric oxide (NO) production [52]. Studies have confirmed that exposure to cell-free hemoglobin significantly reduces adropin expression in vascular endothelial cells, suggesting that adropin could be an important predictive marker for the occurrence of endothelial cell damage after craniotomy in the presence of cell-free hemoglobin [50]. In addition, Banerjee et al. found in mice that adropin was significantly expressed in neurons, oligodendrocyte progenitor cells, oligodendrocytes cells, and microglia, and its expression correlated with neurodegeneration and cell metabolism-related genomes. Treatment with synthetic adropin peptide also reversed age-related cognitive decline and affected the expression of genes involved in morphogenesis and cell metabolism, suggesting an important role for adropin in brain cognitive function [53]. The expression of adropin can be further investigated in the future after MVD surgery and may be able to serve as an important marker of brain cognitive function impairment.

Although delirium is thought to be distinct from PND, the two syndromes may be highly correlated in the short and long term [54–57]. Various studies have shown that delirium might accelerate the cognitive decline in patients with Alzheimer's disease [58–60]. In this study, results showed that patients with POD were more likely to develop PND than patients without POD. Therefore, delirium may have long-term mental health complications that have not been fully investigated and may affect neurofunctional recovery. A larger scale of epidemiological studies will be necessary to identify the relationship between the occurrence of delirium and PND.

Despite the positive results of this study, it had several limitations. Peripheral blood samples were obtained in the postoperative period, meaning that delirium did not formally precede their collection. It is, therefore, possible that delirium contributed to changes in postoperative serum biomarkers. Secondly, 32 patients failed to follow up, and the relationship between serum biomarkers and PND could not be adequately analyzed. Furthermore, due to the fact that all patients in this study were operated on by the same senior neurosurgeon, the model may not be generalizable to other surgical specialties. Therefore, multicenter studies and standard methods of sample collection, as well as long-term follow-up of patients, are necessary to validate these preliminary results in the future.

## 5. Conclusions

This study demonstrated that postoperative serum NSE levels are independently correlated with serum S100β levels and associated with the development of POD. Additionally, serum NSE levels provide a better indication of POD risk in older adults who undergo MVD surgery. Hence, in older patients who had MVD surgery, NSE might play a role in the pathophysiology of POD, and its clinical determination might help identify those at risk for POD.

**Author Contributions:** Conceptualization, J.Q.; data curation, T.G., Z.L. and Z.W.; formal analysis, T.G. and Z.L.; funding acquisition, Z.W.; investigation, J.Q.; methodology, Z.L.; project administration, J.Q. and Z.W.; resources, T.G. and Z.L.; software, Z.L.; supervision, J.Q. and Z.W.; validation, Z.W.; visualization, Z.W.; writing—original draft, T.G.; writing—review and editing, J.Q. and Z.W. All authors have read and agreed to the published version of the manuscript.

**Funding:** This work was supported by the National Natural Science Foundation of China (Grant Nos. 62027813 and 8180100922), the Beijing Municipal Science and Technology Commission (Grant No. 7192056), and the Natural Science Foundation of Beijing (Grant No. J180005).

**Institutional Review Board Statement:** The study was conducted in accordance with the Declaration of Helsinki, and approved by the Institutional Review Board of Beijing Tiantan Hospital, Capital Medical University (protocol code KY2019-022-02, 16 April 2019).

**Informed Consent Statement:** Informed consent was obtained from all participants included in this study. Written informed consent has been obtained from the patient(s) to publish this paper.

**Data Availability Statement:** Not applicable.

**Conflicts of Interest:** The authors declare no conflict of interest.

## Abbreviations

POD: postoperative delirium; MVD: microvascular decompression; NSE: neuro-specific enolase; PND: perioperative neurocognitive disorders; HFS: hemifacial spasm; TN: trigeminal neuralgia; MMSE: Mini-Mental State Examination; CAM: Confusion Assessment Method; DSM-V: Diagnostic and Statistical Manual of Mental Disorders; MDAS: Memorial Delirium Assessment Scale; OR: odds ratio; CI: confidence interval; AUC: area under the curve; PR: precision–recall curve.

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
