# Peer review of "Relationship of Postoperative Serum Neuro-Specific Enolase Levels with Postoperative Delirium Occurring after Microvascular Depression Surgery in Older Patients"

_2813-2475, doi:10.3390/jvd2010001_

Round 1
Reviewer 1 Report
The Authors present an interesting article aiming at investigating the association between NSE and incident delirium in older patients undergoing MVD.
Some minor grammatical and lexical error should be corrected to increase the readability of the paper.
Please, modify the word “elderly” with “older patients”/”older participant”, as “elderly” is considered offensive by a significant share of older persons.
I don’t clearly understand the study design. The authors state that they included cases (those who underwent MVD) and controls (healthy individuals). However, the outcome of the study was post-operative delirium which, by definition, cannot occur in those who did not undergo MVD. The Authors should clearly state what is the benefit of including healthy controls in this study, rather than showing that pre-operative levels of NSE were similar between the two groups.
The Authors found an incidence of POD of 22%, even if the study population was selected by excluding typical risk factors for delirium (such as sensory impairment, cognitive decline, and multimorbidity). It would be interesting to better characterize the delirium developed by these patients: a descriptive table with the CAM assessment should be reported. This would help the readers to understand the severity and the subtype of delirium developed (hyperactive? Mixed? Hypoactive?)
The Authors state that postoperative NSE levels may help to identify individuals who will develop POD from the others. However,
1) they previously state that there are no recommendations on the best way to reduce the incidence of POD: therefore, what is the benefit of identifying older persons at increased risk of delirium?
2) the results are neither internally or externally validated and further study are needed – this is particularly important in consideration of the low generalizability of the results due to the exclusion criteria
3) the study is characterized by a significant class imbalance (20% of POD vs 80% non-POD): ROC-AUC has been shown to be sensitive to imbalanced class problems. I suggest producing precision-recall curves and calculating the PR-AUC, in order to give to the readers a clear vision of the benefit of implementing post-operative NSE evaluation in clinical practice.
Author Response
1."Some minor grammatical and lexical error should be corrected to increase the readability of the paper.”
Author: We’ve modified the corresponding statement.
2. “Please, modify the word “elderly” with “older patients”/”older participant”, as “elderly” is considered offensive by a significant share of older persons.”
Author: We’ve modified the corresponding statement.
3. “I don’t clearly understand the study design. The authors state that they included cases (those who underwent MVD) and controls (healthy individuals). However, the outcome of the study was post-operative delirium which, by definition, cannot occur in those who did not undergo MVD. The Authors should clearly state what is the benefit of including healthy controls in this study, rather than showing that pre-operative levels of NSE were similar between the two groups.”
Author: The main body of this paper was 209 patients who underwent MVD surgery. The 209 normal controls were introduced only to show that these patients with facial spasms or trigeminal neuralgia had the same NSE levels as healthy individuals and that delirium following MVD surgery was consistent with elevated NSE levels. Thus the inclusion of normal controls can more clearly illustrate the synergistic relationship between changes in NSE levels and the occurrence of delirium.
4. “The Authors found an incidence of POD of 22%, even if the study population was selected by excluding typical risk factors for delirium (such as sensory impairment, cognitive decline, and multimorbidity). It would be interesting to better characterize the delirium developed by these patients: a descriptive table with the CAM assessment should be reported. This would help the readers to understand the severity and the subtype of delirium developed (hyperactive? Mixed? Hypoactive?)”
Author: It is indeed important for the further evaluation of delirium and we appreciate your advice deeply. We classified 46 patients who developed delirium after surgery into hyperactive, hypoactive, and mixed types, and the severity of delirium was assessed using the MDAS(Memorial Delirium Assessment Scale). Then, we compared the differences in postoperative serum NSE expression levels between different types of delirium and evaluated the correlation between postoperative serum NSE levels and MDAS scores.
5. “1) they previously state that there are no recommendations on the best way to reduce the incidence of POD: therefore, what is the benefit of identifying older persons at increased risk of delirium?”
Author: Most of the patients after MVD are generally in good condition, but if delirium occurs in some elderly patients, it affects the patients' ability to take care of themselves to some extent, and even unexpected events such as falls and impulsive behaviors occur. For the related high-risk patients, patients can be given a higher level of care and attention, and family members can accompany the bed if necessary to avoid related accidents. In addition, the discovery of intrinsic relationships between clinical indicators can lay the foundation for subsequent in-depth mechanistic studies.
6. “2) the results are neither internally or externally validated and further study are needed – this is particularly important in consideration of the low generalizability of the results due to the exclusion criteria”
Author: Validating internally and externally is indeed very beneficial for both model building and validation. However, since the incidence of postoperative delirium was not very high, only 46 cases in our study and if the samples were divided into two groups for validating internally and externally, the model building and validation would be less effective due to the reduced number of cases, which is not conducive to screening for the most valuable clinical predictive indicators. More cases could be included for further validation of the results in the future.
7. “3) the study is characterized by a significant class imbalance (20% of POD vs 80% non-POD): ROC-AUC has been shown to be sensitive to imbalanced class problems. I suggest producing precision-recall curves and calculating the PR-AUC, in order to give to the readers a clear vision of the benefit of implementing post-operative NSE evaluation in clinical practice.”
Author: Thank you very much for your suggestions. The PR curves were plotted in R studio and the associated AUCs were calculated. The results are presented in Figure 3.
Reviewer 2 Report
Interesting paper looking NSE in relation to delirium post MVD. The paper is well designed and of interest.
Should expand the discussion and specifically focus on adropin as additional biomarker PMID: 34710824.
If the above concepts and reference included, could be of interest to broader audience.
Author Response
Thank you for your kind suggestions. We have refined the discussion section by adding thoughts on the mechanisms involved in the occurrence of delirium and the arguments for adropin as a potential marker.
Round 2
Reviewer 2 Report
Accept
Author Response
Thank you very much for your kind comments. I hope everything goes well for you.